# Prognostic Implication of Exfoliative Airway Pathology in Cancer-Free Coal Workers’ Pneumoconiosis

**DOI:** 10.3390/ijerph192214975

**Published:** 2022-11-14

**Authors:** Uiju Cho, Tae-Eun Kim, Chan Kwon Park, Hyoung-Kyu Yoon, Young Jo Sa, Hyo-Lim Kim, Tae-Jung Kim

**Affiliations:** 1Department of Hospital Pathology, St. Vincent Hospital, The Catholic University of Korea, 93 Jungbu-daero, Suwon 16247, Korea; 2Department of Hospital Pathology, Yeouido St. Mary’s Hospital, College of Medicine, The Catholic University of Korea, 10, 63-ro, Seoul 07345, Korea; 3Division of Pulmonary and Critical Care Medicine, Department of Internal Medicine, Yeouido St. Mary’s Hospital, College of Medicine, The Catholic University of Korea, Seoul 07345, Korea; 4Department of Thoracic Surgery, Yeouido St. Mary’s Hospital, College of Medicine, The Catholic University of Korea, 10, 63-ro, Seoul 07345, Korea; 5Department of Radiology, Yeouido St. Mary’s Hospital, College of Medicine, The Catholic University of Korea, 10, 63-ro, Seoul 07345, Korea

**Keywords:** pneumoconiosis, alveolitis, progressive massive fibrosis, coal worker, miner, cytology, bronchial washing, goblet cell hyperplasia, squamous metaplasia

## Abstract

Background: The purpose of this study is to see if exfoliative pulmonary airway pathology in cancer-free coal workers’ pneumoconiosis (CWP) can be used as a biomarker for predicting pulmonary morbidity. Methods: We investigated persistent metaplastic changes in bronchoscopic washing cytology and differential cell counts in bronchoalveolar lavages (BAL) in 97 miners with CWP and 80 miners without CWP as the control. Clinicopathological parameters were examined including pulmonary function tests and the presence of progressive massive fibrosis. Results: When compared to the control group, severe alveolitis, severe goblet cell hyperplasia (GCH), severe hyperplastic epithelial change, and severe squamous metaplasia were the distinguishing biomarkers in CWP. Multivariate analysis revealed that severe alveolitis and severe GCH, along with miner duration and current smoker, were independent predictors of pulmonary mortality. The survival analysis revealed a significantly different survival rate between the three groups: no evidence of severe alveolitis and severe GCH, presence of severe alveolitis or severe GCH but not both, and both severe alveolitis and severe GCH. Conclusions: The severities of alveolitis and goblet cell hyperplasia in the bronchoscopic study are independent prognostic factors for CWP. A pathologic grading system based on these two parameters could be used in the stratification and clinical management of CWP patients.

## 1. Introduction

Coal workers’ pneumoconiosis (CWP) is a non-neoplastic reaction of the lungs to inhaled mineral or organic dust that does not include asthma, bronchitis, or emphysema. Coal workers’ pneumoconiosis (CWP) is a pulmonary reaction to inhaled coal dust that can take both simple and complicated forms. Many people with CWP are asymptomatic. However, by the time the disease manifests clinically, the lung has suffered extensive irreparable damage. Even if the patient is no longer exposed to coal dust, respiratory disability progresses. Patients have functional deficits in the lungs, including obstructive defects, abnormal diffusing capacities, and restrictive defects. Coal dust exposure causes a variety of co-morbid pulmonary diseases including chronic obstructive pulmonary disease (COPD) [1,2,3]. Decreased pulmonary function in CWP patients contributes to increased premature mortality [4,5,6,7]. Age-adjusted CWP death rates in the United States have been declining since the 1960s, but an increasing trend in the years of potential life lost (YPLL) before the age of 65 has been observed since 2002 [4]. The continued occurrence of CWP-attributable YPLL emphasizes the importance of properly assessing CWP severity and progression risk. The findings of a prospective cohort study of coal miners in the United States confirmed that exposure to coal mine dust increases mortality, with pneumoconiosis having the highest standardized mortality ratio [8]. Patients with CWP frequently displayed respiratory symptoms and a decline in lung function. It has been demonstrated that CWP increases the chance of developing chronic obstructive pulmonary diseases including emphysema and chronic bronchitis [1,9,10]. Chest radiographs are frequently used for pulmonary complication screening or evaluation. However, there are some restrictions on the sensitivity and specificity of imaging workups that begin with simple chest radiographs [11]. The International Labour Office (ILO) categorization did not correlate with the forced expiratory volume in one second (FEV1) or dyspnea in a cross-sectional investigation [12]. High-resolution computed tomography scans showed the strongest association between the severity of the impairment and the emphysema score. Since irreversible lung fibrosis appears to be fed by ongoing inflammation, it was proposed that the measurement of coal-dust-induced inflammatory cytokines could be used as a marker of CWP for the identification of high- and low-risk groups [13]. In therapeutic contexts, however, it is not always easy to get there. Therefore, it could be helpful to have a variety of tools for evaluating respiratory dysfunction and prognosis.

Airway evaluation, including bronchoalveolar lavage (BAL) and bronchial washing, is performed in many CWP patients for various purposes. It is used (1) to find putative mineral dust [14], (2) to assess other causes of lung disease such pneumonia, sarcoidosis, allergic alveolitis, or lung cancer, and occasionally (3) to reduce the mineral dust burden [15]. Numerous past studies have shown that BAL is a simple, safe method for determining whether the lower respiratory tract is inflamed. It has been shown that the type of inflammatory response and the severity of the alveolar structure in interstitial lung illnesses are both reflected by an elevated percentage of lymphocytes and neutrophils in BAL fluid, which is frequently referred to as “alveolitis” [16,17]. Interstitial pulmonary fibrosis patients are at a higher risk of deterioration when they have a higher neutrophil count [18]. Occupational lung diseases have drawn attention to BAL fluid as well. Previous research on coal miners has demonstrated that the BAL fluid contains elevated levels of alveolar macrophages, which release excessive reactive oxygen species, several cytokines, and surfactant proteins [19,20]. Unlike other interstitial lung diseases, the usefulness of BAL in relation to disease progression and prognosis remains unclear in CWP. By washing the bronchi, it is possible to sample inflammatory cells, caustic organisms, and bronchial epithelial cells in the event of an infection. Certain bronchial epithelial cell characteristics, including squamous metaplasia, hyperplastic epithelial change, and goblet cell hyperplasia, reflect the outcomes of abnormal airway repairs from acute and chronic injuries. However, the clinical ramifications of the continued discovery of these changes in the bronchial epithelial cell in pneumoconiosis patients need to be explored. We postulated that a long-standing inflammatory status of the distal air spaces that is objectively reflected by BAL fluid and proximal air spaces by bronchial cytology and airway cytology from CWP patients shares similar features to those found in chronic obstructive pulmonary diseases. Additionally, in order to address the role of airway cytology in pulmonary mortality, we looked into whether the characteristics of bronchial airway cytology obtained from CWP, when compared with radiologic and functional parameters, have predictive value for the prognosis and short-term survival in cancer-free CWP patients.

## 2. Materials and Methods

### 2.1. Subjects

Between January 2000 and 31 December 2015, 4545 consecutive retired coal miners were admitted as examinees for CWP diagnosis at the Occupational and Environmental Medicine Center and the Department of Pulmonology. CWP was diagnosed in 2544 miners (based on ILO classification, profusion of small, irregular opacities of grade 1/0 or greater were present). Among them, 179 patients with CWP had a clinical follow-up in routine chest radiography, pulmonary function test (PFT), and bronchoscopy cytology including bronchial washing and BAL due to infections, pneumonia, and localized lung lesions. Of these patients, 82 were not included in the study because they had a type of malignancy both at the time of the bronchoscopy and during the follow-up. The subjects of this study were the remaining 97 CWP patients (mean age, 67 years; 71 men, 26 women; 51 smokers, 26 ex-smokers, and 20 non-smokers). Among the 143 retired miners without CWP (ILO classification of 0/0 or 0/1) who underwent bronchoscopy examinations during the investigation of localized pulmonary lesions between 1 January 2008 and 31 December 2011, 80 had available bronchoscopic cytology samples without any evidence of malignancy and were used as controls (61 males and 19 females; 36 smokers, 25 ex-smokers, and 19 non-smokers; mean age of 65 years) (Figure 1). The subjects were classified as current smokers, former smokers, or never smokers. To qualify as a former or current smoker, one had to have smoked more than 100 cigarettes. By the end of 2018, a CWP-related death had been determined and the date was used to determine the number of months of follow-up. The Institutional Review Boards approved this study.

### 2.2. Pulmonary Function Test

The pulmonary function test (PFT) was carried out in accordance with the published standards of the American Thoracic Society [21]. Subjects were grouped into four patterns: (1) Normal: FVC > 80%, and FEV1/FVC ≥ 70%; (2) Obstructive: FVC > 80% and FEV1/FVC < 70%; (3) Restrictive: FVC ≤ 80% and FEV1/FVC ≥ 70%; and (4) Mixed: FVC ≤ 80% predicted and FEV1/FVC < 70% [22]. Restrictive, obstructive, and mixed patterns were defined as abnormal PFT.

### 2.3. Chest Radiography and International Labor Organization Scores

Chest radiographs were reviewed by an experienced panel of physicians and radiologists and scored according to the classification rules of the International Labor Organization (ILO) 2002 classification system [23]. Accordingly, patients were divided into simple CWP and complicated CWP.

### 2.4. Bronchoalveolar Lavage Analysis

BAL was performed as part of routine clinical management following the recommended guidelines [24,25]. BAL was carried out in one of the subsegmental bronchi of the right-middle lobe using 0.9% saline with four successive aliquots of 50 mL using a wedged flexible fibroptic bronchoscope (BF-B3R; Olympus, Tokyo, Japan). The fluid returned from the first 50 mL aliquot was not used because it contained known cells and material coming from airways larger than the peripheral regions [26]. Differential cell count was performed in a routine process. The volume of recovered BAL fluid was assessed after filtering it from mucus with a Dacron net (Millipore, Cork, Ireland). Both the absolute volume and percentage of injected fluid were used to calculate the recovery. The cells were pelleted by centrifugation at 400× *g* for 10 min at 4 °C and the supernatants were collected. The cell pellets were resuspended in an RPMI medium and the total cell counts were performed in a Burker chamber (Marienfeld, Germany). From the results of the differential cell count, alveolitis was defined as a proportion of neutrophils ≥3% and/or eosinophils ≥2% on a BAL based on previous studies on systemic sclerosis and the healthy human lung [27,28]. Furthermore, the severity of alveolitis was subdivided by the percentage of inflammatory cells: (1) absent: neutrophils <3% and eosinophils <2%; (2) mild: neutrophils ≥3% and <5%, and eosinophils ≥2% and <3%; (3) moderate: neutrophils ≥5% and <7%, and eosinophils ≥2% and <3%; (4) severe: neutrophils ≥7% and/or eosinophils ≥3%.

### 2.5. Bronchial Washing Cytology Analysis

All the bronchial washing slides of the CWP patients and control group were reviewed by two pathologists. The pathologists were blinded to all information on the cases.

Then, we graded those cytology findings and reviewed them again for all subjects. Then, we compared the CWP and control groups and identified the specific findings that distinguished the CWP group from the control group. Non-neoplastic cytological features of the exfoliative cells, including goblet cell metaplasia, hyperplastic epithelial change, and squamous metaplasia, were examined and histologically defined as follows. The goblet cell was an oblong to columnar, non-ciliated cell, with a basally located, round nucleus, with finely granular chromatin, micro-nucleoli, and an abundant pale-lacy-pink cytoplasm, which was found by positive periodic acid-Schiff staining [29]. The presence of more than 50% goblet cells in each bronchial epithelial cluster was characterized as goblet cell hyperplasia. Hyperplastic epithelial alteration was classified as exfoliated cell clusters with uneven, jagged edges, as well as a pseudostratified respiratory epithelium with a syncytial pattern, variably expanded nuclei with evenly dispersed, finely granular chromatin, and conspicuous nucleoli. [29]. Squamous metaplasia was defined as medium-sized, round to polygonal squamous cells, with a dense cytoplasm and a low nucleus/cytoplasm ratio [29]. An alveolar dust-laden macrophage was defined as a cluster of round to oval cells with a round nucleus and phagocytic cytoplasm [29]. The degrees of the cytologic findings were graded semi-quantitatively based on the proportion of abnormal cytologic features in all exfoliated cells examined in whole cytology slides: (1) minimal; <25%, (2) mild, ≥25% and <50%, (3) moderate, ≥50% and ≤75%, (4) severe, >75%.

### 2.6. Statistical Analysis

Statistical analyses were performed using GraphPad prism for macOS version 9 statistical software (San Diego, CA, USA). All the associations between the categorical variables were performed using the Chi-square exact test and Fisher exact test (two-sided). The comparison of the means of two independent variables was analyzed by the Mann–Whitney test. The prognostic values of the independent factors were examined in relation to survival using the Kaplan–Meier analysis or Mantel–Cox test for the univariate analysis and Cox proportional-hazards regression for the multivariate analysis. For the survival curve comparison of three exfoliative pathology grades, the log-rank (Mantel–Cox) test was used; p values less than 0.05 were considered statistically significant. Data are presented as means and standard deviations.

## 3. Results

### 3.1. Bronchoalveolar Lavage Differentials and Exfoliative Cytology Findings

The absolute numbers of macrophages, neutrophils, lymphocytes, and eosinophils in the BAL fluid of patients with CWP are demonstrated in Table 1. Because of a skewed distribution, the absolute cell numbers in the BAL fluid are expressed as median values, with the interquartile ranges in parentheses. The CWP group’s differential cell profile was dominated by alveolar macrophages (86.9 ± 5.2%). Lymphocytes, neutrophils, and eosinophils accounted for 4.8 ± 2.7%, 6.5 ± 3.5%, and 1.8 ± 1.5%, respectively. The control group’s cell profile included 89.8 ± 3.5% alveolar macrophages, 4.6 ± 2.7% lymphocytes, 4.7 ± 1.7% neutrophils, and 0.9 ± 0.6% eosinophils. The means for the percentages of neutrophils and eosinophils in the CWP group were significantly higher than in the control group (p < 0.001 and p < 0.001, respectively). The mean of macrophages was significantly lower in the CWP group than in the control group (p < 0.001), but there was no statistical difference between the two groups for lymphocytes (p = 0.915) (Figure 2). The differential count for the BAL was observed to be significantly different between the CWP and control groups, except for the lymphocyte count. The absolute numbers of goblet cell hyperplasia, hyperplastic change, and squamous metaplasia are demonstrated in Table 2. In Table 3, the cytologic results of the bronchial washing cytology are shown, as well as alveolitis from the differential cell count for the BAL. In contrast to the controls, severe alveolitis (p = 0.001), severe goblet cell hyperplasia (p < 0.001), hyperplastic epithelial change (p = 0.023), and squamous metaplasia (p < 0.001) were more prevalent in the CWP group.

The distribution of the differential count for the BAL according to various clinical factors within the CWP group was next examined; No significant difference between current and never or former smokers was found. The percentage cell counts of macrophages and neutrophils were significantly different between the miner duration (≥25 versus <25 yr), ILO classification (complicated versus simple), and pulmonary function test (abnormal versus normal) (Figure 3).

### 3.2. Patient Demographics and Pulmonary Exfoliative Cytology Findings

Table 4 displays the patient demographics and their associations with the exfoliative cytologic findings. In terms of pulmonary function, 30 (81%) out of the 37 CWP patients with severe alveolitis had abnormal PFT results. A severe degree of goblet cell hyperplasia was also linked to abnormal PFT results. There was no correlation between hyperplastic epithelial change and squamous metaplasia and functional status as measured by a PFT. We then looked at the relationship between severe alveolitis and severe GCH in terms of clinical and radiologic characteristics. Severe alveolitis and GCH were found to be significantly higher in males and complicated CWP. Patients with a longer mining work history had a significantly higher risk of severe alveolitis (*p* = 0.012).

### 3.3. Prognostic Value of Pulmonary Exfoliative Cytology Findings

The mean follow-up duration for 97 patients from the initial bronchoscopic examination was 31 ± 30 months. There were 15 CWP-related pulmonary morbidities in the retrospective review (13 acute respiratory failure and 2 pneumonia). The mean survival period was 31 ± 25 months. The multivariate-adjusted hazard ratios (HR) according to the clinicopathologic parameters are shown in Table 5. We observed a significantly increased risk of mortality for a long miner duration (HR 5.451, *p* = 0.009), a current smoker (HR 5.046, *p* = 0.038), severe alveolitis in the BAL study (HR 5.408, *p* = 0.049), and severe GCH in bronchial cytology (HR 3.532, *p* = 0.039). In contrast, there was no statistically significant increased risk of mortality for the other two patterns: a complicated CWP pattern in a radiologic evaluation and an abnormal PFT.

We carried out additional analyses and divided the patients into three groups by combining the goblet cell hyperplasia and alveolitis groups. Grade 1 included those without severe alveolitis or severe GCH, Grade 2 included those with either severe alveolitis or severe GCH, and Grade 3 included those with both severe alveolitis and severe GCH (Table 6). Survival varied significantly between the three pathologic grades, as shown in Figure 4. The survival rate for Grade 1 remained largely stable, but Grades 2 and 3 had lower survival rates (*p* < 0.001). Within a 5-year follow-up period, the mortality in patients with features of Grade 3 was 58% (7 out of 12).

## 4. Discussion

Although it has been suggested that the cellular alterations in BAL fluid serve as a mirror of lung inflammation in a variety of pneumoconiosis [30,31,32], their clinical utility has yet to be determined. We found that examining the differential cell count in BAL fluid was useful for predicting the prognosis in CWP patients, as well as representing the current inflammatory status. Significantly high BAL neutrophils and/or eosinophils, which defined severe alveolitis, were critical. Previous BAL study data for CWP are contentious. Rom and colleagues discovered that alveolar macrophages dominated the inflammatory cells in nonsmokers with complicated CWP [33]. Although neutrophils made up a very small portion of the total population of inflammatory cells, their proportions were higher in pneumoconiosis (mean value 3.3%). Previously, coal miners without CWP had a higher neutrophil count (mean value 3.7%) than miners with CWP. On the other hand, CWP had a significantly lower lymphocyte count [20]. The proportion of lymphocytes was reduced in a diverse group of patients exposed to inorganic or organic dust. Regardless of the presence of pneumoconiosis, the cellular profiles of miners revealed a significant decrease in lymphocytes [30]. The researchers investigated the utility of BAL cellular profiles for monitoring disease progression and the response to treatments for specific interstitial lung diseases. The degree of increase in BAL neutrophils has been linked to disease activity and prognosis in both idiopathic pulmonary fibrosis [34,35,36] and hypersensitivity pneumonitis [37,38]. Increases in the eosinophil proportions and numbers were seen in many of the asbestos-exposed individuals [38] and BAL eosinophilia has been correlated with worse prognosis in idiopathic pulmonary fibrosis [39]. The eosinophil proportion did not have a significant difference in the pneumoconiosis group in Turkish miners [30]. In our study, moderate alveolitis was more common in the control group, but when the alveolitis was severe enough, it was significantly higher in CWP patients. Severe alveolitis was also linked to long-term coal dust exposure and massive pulmonary fibrosis. In clinical terms, it is possible that the severity of alveolitis is more important than the mean value of inflammatory cells. The precise roles of eosinophils and neutrophils require further investigation.

Goblet cells that produce mucus are typically found in the spaces between ciliated columnar bronchial cells. These cells frequently multiply to form a bronchial mucosa with goblet cell hyperplasia in the presence of environmental toxins, chronic inflammation, or irritation, as seen in patients with asthma, chronic bronchitis, or bronchiectasis. In our study, CWP patients had a higher prevalence of severe goblet cell hyperplasia than the control group, as would be expected. Additionally, there was a statistically significant correlation between severe goblet cell hyperplasia and fibrosis and declining lung function. This information backs up our initial theory that COPD and CWP share similar respiratory cytologic characteristics. Additionally, this supports the data from earlier studies on COPD in coal miners [1,40,41,42]. Goblet cell hyperplasia’s function and significance in CWP, however, are not yet well understood. Mucous cell metaplasia results in mucin overproduction, a major condition that contributes to airway obstruction in many chronic airway diseases, including COPD, asthma, and cystic fibrosis [43]. A variety of factors such as glycoprotein [44] as proteases secreted by inflammatory cells [45,46,47] are known to induce mucus hyper-production. Goblet cell hypertrophy and hyperplasia have been observed in the airway epithelium of cigarette smokers [48]. Furthermore, smokers with airflow obstruction have the highest levels of mucin in the epithelium. Notably, in vitro respiratory epithelial cells, tumor necrosis factor (TNF)-α, and interleukin (IL)-1 can induce glycoprotein secretion via ER and MAP kinase signaling [49]. A similar change has also been observed in alveolar macrophages from coal miners with CWP [50,51]. The severity of CWP was found to be directly related to the magnitude of these critical proinflammatory and fibrogenic cytokine releases [33,52]. TNF-α is a protein that is released by alveolar macrophages after they ingest dust particles and is linked to the initiation of fibroblast proliferation and collagen deposition [53]. Based on these findings and the current study, it is possible to hypothesize that similar cytokine pathways involving TNF-α and IL-1 cause goblet cell hyperplasia in CWP. More research is needed to determine the precise role of cytokines in goblet cell hyperplasia and their relationship to pneumoconiosis progression.

This study is novel in that we used two combined airway cytology parameters, alveolitis and goblet cell hyperplasia, to create a pathologic grading system. Aside from the duration of dust exposure, the parameters that were strongly correlated with cancer-free CWP patients were severe alveolitis and goblet cell hyperplasia. After an 80-month follow-up from the initial bronchoscopic cytology examination, Grade 3 patients had a survival rate of 0.2, which was significantly lower than the rate for Grade 2 patients. It implies that patients with CWP who have severe alveolitis and goblet cell hyperplasia are at high risk of pulmonary morbidity. Therefore, we propose that a different follow-up period and therapeutic approach should be taken with patients who have severe alveolitis and goblet cell hyperplasia and ideally with individuals who have either severe alveolitis or goblet cell hyperplasia in their bronchial washing cytology.

However, our study has several limitations. Because this is a retrospective study, the patient populations and follow-up times differed. Furthermore, we were unable to exclude patients who had a suspected infection at the time of the bronchoscopy. Bias due to patient exclusion is also likely to have influenced the study’s findings.

## 5. Conclusions

The severity of alveolitis and the presence of goblet cell hyperplasia appear to be the two most important predictors of CWP prognosis. Patients who had both severe alveolitis and goblet cell hyperplasia were at the highest risk of dying prematurely. Our findings offer a preliminary guideline for interpreting BAL fluid differential cell count and goblet cell hyperplasia. Pathologic grading in pulmonary morbidity risk bronchoscopic airway cytology may aid in patient management planning. This method of stratifying cytologic phenotypes in CWP could be clinically useful but further research is needed.

## Figures and Tables

**Figure 1 ijerph-19-14975-f001:**
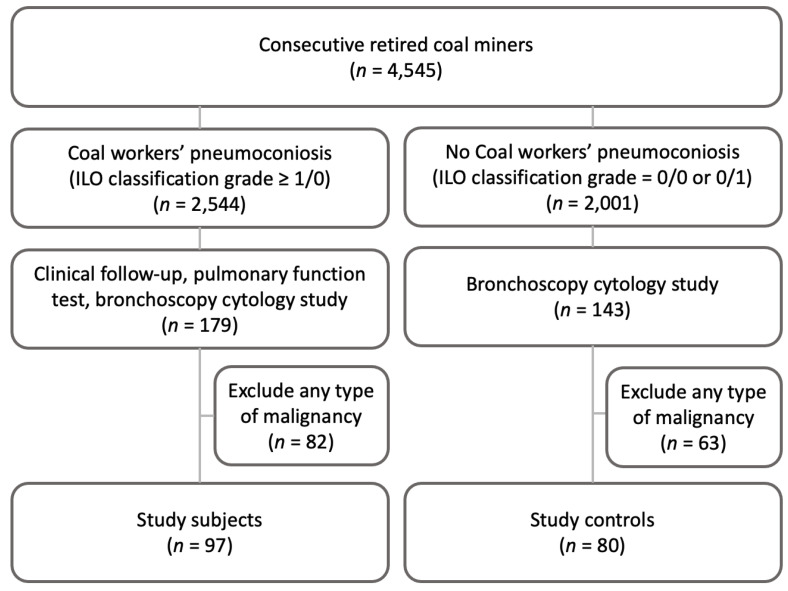
Schematic overview of the study subjects.

**Figure 2 ijerph-19-14975-f002:**
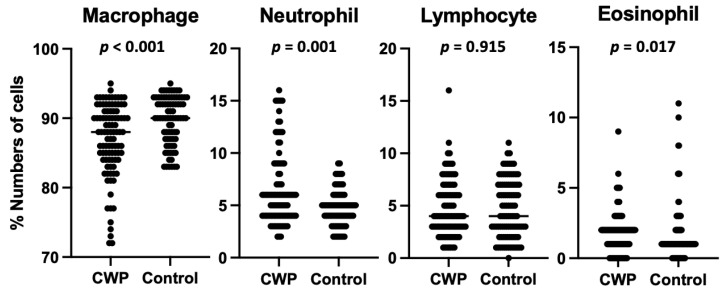
The comparison of bronchoalveolar lavage differential count in coal workers’ pneumoconiosis and controls. The statistical comparison of the mean of the two groups was performed with the Mann–Whitney test. Abbreviation: CWP, coal workers’ pneumoconiosis.

**Figure 3 ijerph-19-14975-f003:**
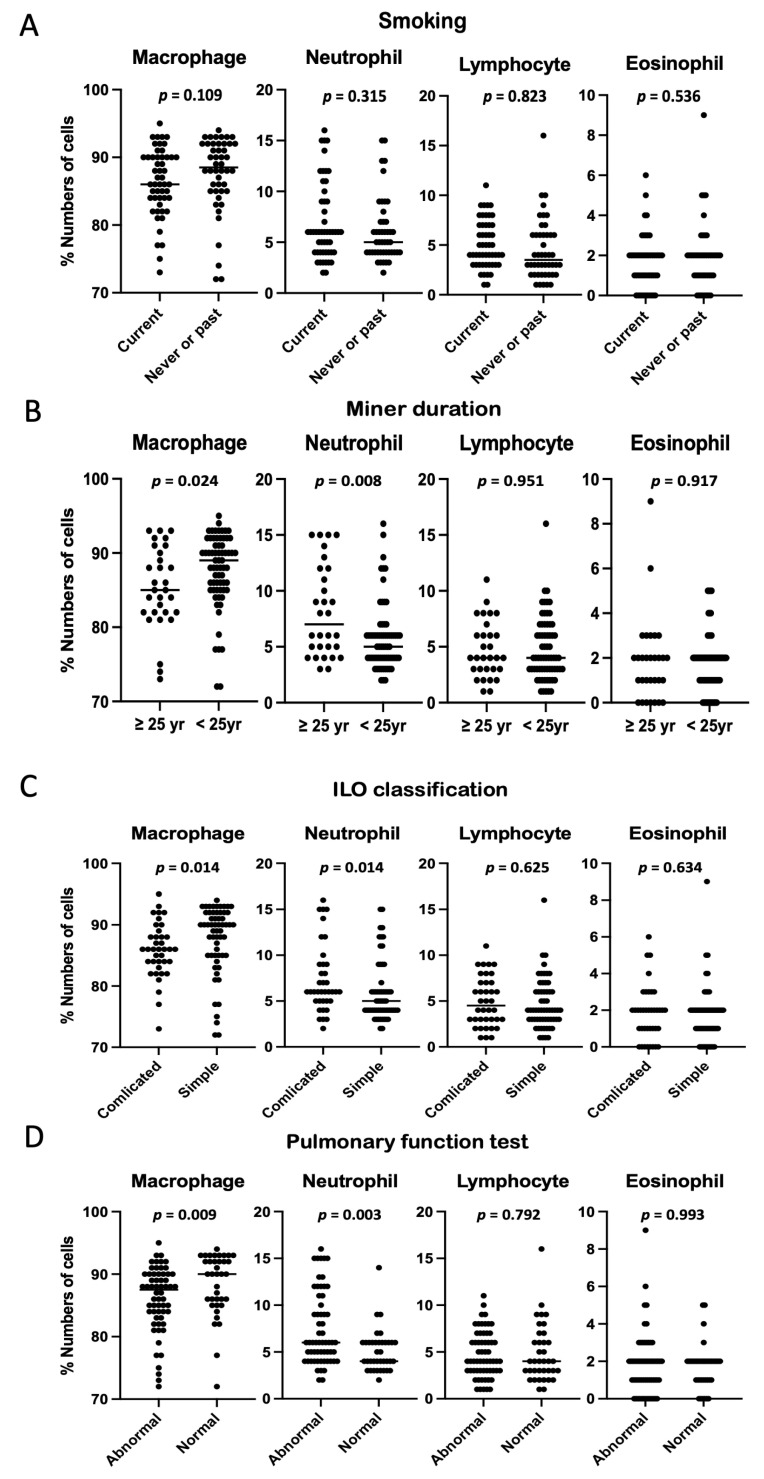
Demonstration of bronchoalveolar lavage differential count in coal workers’ pneumoconiosis based on (**A**) smoking, (**B**) miner duration, (**C**) ILO classification, and (**D**) pulmonary function test. The statistical comparison of the mean of the two groups was performed with the Mann–Whitney test.

**Figure 4 ijerph-19-14975-f004:**
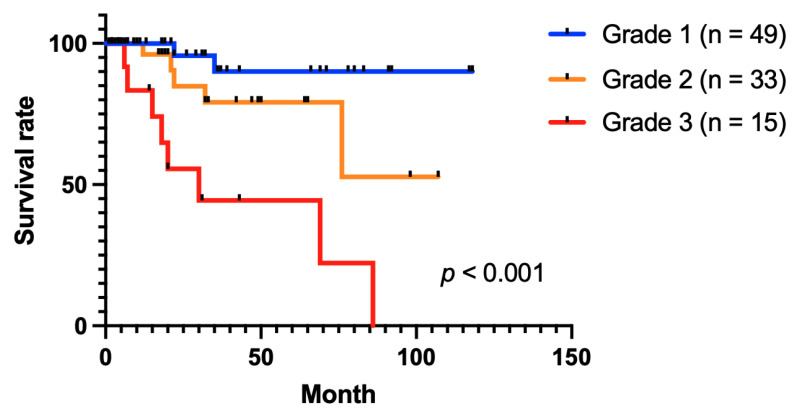
Log-rank (Mantel–Cox) test of patients with coal workers’ pneumoconiosis according to the three grading systems of pulmonary exfoliative cytology.

**Table 1 ijerph-19-14975-t001:** Absolute numbers of cells in bronchoalveolar lavages obtained from patients with coal workers’ pneumoconiosis (n = 97) and control subjects (n = 80).

	Current Smoker		Never or Former Smoker	
	CWP	Control	CWP	Control
n	51	36	46	44
Total cell count	43.5 (28.7–58.3)	23.0 (11.4–29.0)	19.6 (10.5–29.0)	8.8 (7.6–11.2)
Macrophage	30.6 (12.4–48.7)	21.5 (11.2–25.3)	12.8 (7.4–17.9)	7.9 (6.7–9.3)
Neutrophil	1.9 (0.7–7.4)	0.1 (0.3–0.3)	1.4 (0.4–2.9)	0.1 (0.0–0.2)
Lymphocyte	0.7 (0.3–2.3)	0.9 (0.5–1.2)	0.8 (0.5–2.7)	1.0 (0.5–1.3)
Eosinophil	1.2 (0.3–3.1)	0.1 (0.0–0.1)	0.8 (0.1–2.5)	0.0 (0.0–0.0)

Data are expressed as median absolute numbers of cells × 10^4^/mL, with the interquartile ranges in parentheses.
Abbreviation: CWP, coal workers’ pneumoconiosis.

**Table 2 ijerph-19-14975-t002:** Absolute numbers from bronchoscopy exfoliative cytology of coal workers’ pneumoconiosis (n = 97) and controls (n = 80).

	CWP	Control
n	97	80
Total cell count	1048 (445–1654)	859 (343–1284)
Goblet cell hyperplasia	356 (230–925)	174 (84–370)
Hyperplastic change	366 (145–548)	253 (123–421)
Squamous metaplasia	252 (123–445)	434 (132–522)

Data are expressed as median absolute numbers of cells with the interquartile ranges in parentheses. Abbreviation:
CWP, coal workers’ pneumoconiosis.

**Table 3 ijerph-19-14975-t003:** Pathologic findings of bronchoalveolar lavage and bronchoscopy exfoliative cytology from coal workers’ pneumoconiosis.

	Total (n = 177)	Control (n = 80)	CWP (n = 97)	*p*
Alveolitis (%)				<0.001
severe	50 (28)	13 (26)	37 (74)	
mild to moderate	127 (72)	67 (53)	60 (47)	
Goblet cell hyperplasia (%)				<0.001
severe	30 (17)	4 (13)	26 (87)	
mild to moderate	147 (83)	76 (52)	71 (48)	
Hyperplastic change (%)				0.023
severe	48 (27)	15 (31)	33 (69)	
mild to moderate	129 (73)	65 (50)	64 (50)	
Squamoous metaplasia (%)				<0.001
severe	59 (33)	11 (19)	48 (81)	
mild to moderate	118 (67)	69 (58)	49 (42)	

Statistical analysis was based on Chi-squared tests.

**Table 4 ijerph-19-14975-t004:** Correlation between patient demographics and pulmonary exfoliative cytology findings in coal workers’ pneumoconiosis.

		Bronchoalveolar Lavage		Bronchial Exfoliative Cytology					
Parameters	N (%)	Severe Alveolitis	*p*	Severe GCH	*p*	Severe HC	*p*	Severe SM	*p*
Age, yr	67 ± 8	65 ± 10		66 ± 10					
<70 yr	50 (51)	44 (59.5)	0.083	13 (50.0)	0.854	20 (60.6)	0.2	22 (45.8)	0.265
≥70 yr	47 (49)	30 (40.5)		13 (50.0)		13 (39.4)		26 (54.2)	
Gender									
Male	71 (73)	32 (86.5)	0.032	23 (88.5)	0.043	25 (75.8)	0.683	38 (79.2)	0.189
Female	26 (27)	5 (13.5)		3 (11.5)		8 (24.2)		10 (20.8)	
Miner duration	18 ± 9	20 ± 10		21 ± 10					
<25 yr	67 (69)	20 (54.1)	0.012	15 (57.7)	0.142	22 (66.7)	0.713	34 (70.8)	0.71
≥25 yr	30 (31)	17 (45.9)		11 (42.3)		11 (33.3)		14 (29.2)	
Smoking									
Never or former	46 (47)	16 (43.2)	0.517	9 (34.6)	0.126	16 (48.5)	0.88	22 (45.8)	0.756
Current	51 (53)	21 (56.8)		17 (65.4)		17 (51.5)		26 (54.2)	
Tuberculosis Hx									
Absent	64 (66)	26 (70.3)	0.484	16 (25)	0.576	20 (60.6)	0.423	31 (64.6)	0.774
Present	33 (34)	11 (29.7)		10 (30)		13 (39.4)		17 (35.4)	
PFT									
Normal	37 (38.1)	7 (18.9)	0.002	4 (15.4)	0.005	10 (30.3)	0.254	18 (37.5)	0.897
Abnormal	60 (61.9)	30 (81.1)		22 (84.6)		23 (69.7)		30 (62.5)	
ILO class									
Simple									
Category 1	23 (23)	7 (18.9)	0.018 ^*a*^	4 (15.4)	0.006 ^*a*^	10 (30.3)	0.009 ^*a*^	12 (25.0)	0.215 ^*a*^
Category 2	26 (27)	7 (18.9)		4 (15.4)		12 (36.4)		12 (25.0)	
Category 3	10 (10)	3 (8.1)		2 (7.7)		4 (12.1)		2 (4.2)	
Complicated									
Type A	14 (15)	5 (13.5)		4 (15.4)		1 (3.0)		5 (10.4)	
Type B	14 (15)	10 (27.0)		6 (23.1)		3 (9.1)		10 (20.8)	
Type C	10 (10)	5 (13.5)		6 (23.1)		3 (9.1)		7 (14.6)	

Data are presented as mean (standard deviation) or number (%). Statistical analysis was based on Chi-squared
tests. ^*a*^ Simple CWP versus complicated CWP. Abbreviations: GCH, goblet cell hyperplasia; HC, hyperplastic
change; SM, squamous metaplasia; PFT, pulmonary function test.

**Table 5 ijerph-19-14975-t005:** Survival analysis of pulmonary morbidity of cancer-free coal workers’ pneumoconiosis.

	Univariate	Multivariate		
Factors	*p*	HR	95%CI	*p*
Age (≥70 yr)	0.627			
Gender (male)	0.108			
Miner duration (≥25 yr)	0.002	5.451	1.595–21.34	0.009
Current smoker	0.016	5.046	1.327–33.24	0.038
Tuberculosis history	0.154			
Complicated CWP	0.022	1.751	0.499–7.018	0.392
Abnormal PFT	0.033	2.475	0.427–20.57	0.342
Severe alveolitis	0.002	5.408	0.963–17.17	0.049
Severe GCH	<0.001	3.532	1.116–12.92	0.039
Severe HC	0.849			
Severe SM	0.347			

Statistical analysis was performed with the Kaplan–Meier analysis for univariate analysis and the Cox proportionalhazards regression for multivariate analysis. Abbreviations: GCH, goblet cell hyperplasia; HC, hyperplastic change; SM, squamous metaplasia; PFT, pulmonary function test.

**Table 6 ijerph-19-14975-t006:** Stratification of pulmonary morbidity of cancer-free coal workers’ pneumoconiosis patients by bronchoscopic exfoliative cytology.

Risk Group *p*	Material	Findings
Grade 1	BAL differential count	neutrophils <7.0% and/or eosinophils <3.0%
	Bronchoscopic cytology	absence of GCH in >90% of all bronchial epithelial clusters
Grade 2	BAL differential count	neutrophils <7.0% and/or eosinophils <3.0%
	Bronchoscopic cytology	GCH >90% of all bronchial epithelial clusters
		or
	BAL differential count	neutrophils ≥7.0% and/or eosinophils ≥3.0%
	Bronchoscopic cytology	absence of GCH in >90% of all bronchial epithelial clusters
Grade 3	BAL differential count	neutrophils ≥7.0% and/or eosinophils ≥3.0%
	Bronchoscopic cytology	and GCH >90% of all bronchial epithelial clusters

Abbreviations: BAL, bronchoalveolar lavage; GCH, goblet cell hyperplasia.

## Data Availability

Not applicable.

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
