# Peer review of "Prognostic Implication of Exfoliative Airway Pathology in Cancer-Free Coal Workers’ Pneumoconiosis"

_ijerph, 2022, doi:10.3390/ijerph192214975_

Round 1

Reviewer 1 Report

Material and Methods

Page 4 lines 112-121

I consider it important that researchers describe the method used for cell counting and describe the number of cells per mm3 or mL they found (either the mean or median) and anatomical sites and expressed in absolute numbers  because if they only represent percentages, these are subjective and do not necessarily represent an inflammatory state.

RESULTS

Page 5 figure 2

Why are there 2 graphs for lymphocytes with different p-values and neutrophils are not included?

Page 6 Table 2

Table 2. Specify which statistical test was used to obtain the p-value for each data series. 

Page 6 lines 181-182

Reviewing this data, the figures make me assume a very wide variance which would apply to a calculation of the median and minimum and maximum ranges. However, the nature of the distribution of the data can be tested by a statistical test of your choice.

Similar comments in lines 183-185

Table 3

Describe tests used to calculate p-values since the authors are assuming a normal distribution throughout the paper, however the variations shown do not demonstrate this.

table 4.

I would like to be able to see the absolute count of each cell based on the total cell count.

Discussion Page 9 lines 240-249

These references are superfluous since they focus on the synthesis and pathophysiology of MUC5B, which was not addressed in this work. I would therefore recommend its omission. 

Author Response

Material and Methods

Page 4 lines 112-121

I consider it important that researchers describe the method used for cell counting and describe the number of cells per mm3 or mL they found (either the mean or median) and anatomical sites and expressed in absolute numbers  because if they only represent percentages, these are subjective and do not necessarily represent an inflammatory state.

Thank you for the precious comment. We added absolute number of cells (table1 and 2) and detailed method of BAL procedure.

RESULTS

Page 5 figure 2

Why are there 2 graphs for lymphocytes with different p-values and neutrophils are not included?

Thank you for the precious comment. We corrected the graphs

Page 6 Table 2

Table 2. Specify which statistical test was used to obtain the p-value for each data series. 

Thank you for the precious comment. We added statistical test in every table

Page 6 lines 181-182

Reviewing this data, the figures make me assume a very wide variance which would apply to a calculation of the median and minimum and maximum ranges. However, the nature of the distribution of the data can be tested by a statistical test of your choice.

Thank you very much for the very important comment. We re-analyzed the data with appropriate statistical method for skewed distribution and described them all.

Similar comments in lines 183-185

Table 3

Describe tests used to calculate p-values since the authors are assuming a normal distribution throughout the paper, however the variations shown do not demonstrate this.

Thank you very much for the very important comment. We re-analyzed the data with appropriate statistical method for skewed distribution and described them all.

table 4.

I would like to be able to see the absolute count of each cell based on the total cell count.

Thank you for the precious comment. We added absolute number at the table 1,2

Discussion Page 9 lines 240-249

These references are superfluous since they focus on the synthesis and pathophysiology of MUC5B, which was not addressed in this work. I would therefore recommend its omission. 

Thank you for the precious comment we omitted content.

Reviewer 2 Report

The manuscript discusses possible prognostic markers of coal workers` pneumoconiosis that could predict morbidity and mortality. The authors concluded through a retrospective study that the most predictive markers regarding 5 years of survival are goblet cell hyperplasia, grade of alveolitis, years spent with coal dust exposure, and smoking. The study points out crucial aspects of the disease and provides valuable guidance to predict outcomes for patients with the disease.

Critics:

·         In the result section from lines 153 to 160, the text describes the cytological finding in BAL for CWP versus the control group. Then from lines 161 to 164, it jumps to explain Figure 2, where the patients’ groups are divided into two new groups regarding pulmonary function (normal and abnormal). This fast change is confusing and hard to understand why neutrophils and eosinophils were significantly different in the case of CWP vs. control grouping and how lymphocytes and macrophages became significantly different in normal vs. abnormal lung function grouping.

To resolve the confusion, I would suggest graphing the cytology findings in CWP vs. control in a similar manner as shown in figure 2 and extending the explanation regarding these two sections.

·         In table 1, The patient numbers do not add up to 80 in the control group regarding the goblet cell hyperplasia (41+72 is not =80 or to the total 30= control+CWP). The number of patients is possible 4 instead of 41.

·         In table 1, please show the p-value according to line 168 in a similar manner as shown in the cases of alveolitis and goblet cell hyperplasia.

·         In Figure 2, two graphs are labeled Lymphocyte. Please correct it.

·         I would suggest adding one paragraph to the discussion which would describe therapeutic choices from the perspective of patients sorted into grades (grade 1 to grade 3).

Minor critics:

·         I believe that in line 24, the phrase irreparable damage should be used instead of reparable.

·         Please delete the repeated partial sentence at line 131 (“and abundant, pale, lacy, pink cytoplasm”)  

·         The alpha signs are missing after TNF- in multiple places in the discussion section (lines 247, 249, 262 and 255) 

Author Response

  • In the result section from lines 153 to 160, the text describes the cytological finding in BAL for CWP versus the control group. Then from lines 161 to 164, it jumps to explain Figure 2, where the patients’ groups are divided into two new groups regarding pulmonary function (normal and abnormal). This fast change is confusing and hard to understand why neutrophils and eosinophils were significantly different in the case of CWP vs. control grouping and how lymphocytes and macrophages became significantly different in normal vs. abnormal lung function grouping.

To resolve the confusion, I would suggest graphing the cytology findings in CWP vs. control in a similar manner as shown in figure 2 and extending the explanation regarding these two sections.

; Thank you for the precious comment. We added table1,2 and Figure 3 to reduce confusion.

  • In table 1, The patient numbers do not add up to 80 in the control group regarding the goblet cell hyperplasia (41+72 is not =80 or to the total 30= control+CWP). The number of patients is possible 4 instead of 41.

; Thank you for the precious comment. We corrected 41 to 4

  • In table 1, please show the p-value according to line 168 in a similar manner as shown in the cases of alveolitis and goblet cell hyperplasia.

; Thank you for the precious comment. We corrected p-value missing.

  • In Figure 2, two graphs are labeled Lymphocyte. Please correct it.

; Thank you for the precious comment. We corrected Figure.

  • I would suggest adding one paragraph to the discussion which would describe therapeutic choices from the perspective of patients sorted into grades (grade 1 to grade 3).

; Thank you for the precious comment. We described the suggestion of different F/U and therapeutic approaches according to cytology grades.

Minor critics:

  • I believe that in line 24, the phrase irreparable damage should be used instead of reparable.

Thank you for the comment. we corrected typo

  • Please delete the repeated partial sentence at line 131 (“and abundant, pale, lacy, pink cytoplasm”)  

 Thank you for the comment. we corrected typo

  • The alpha signs are missing after TNF- in multiple places in the discussion section (lines 247, 249, 262 and 255) 

Thank you for the comment. we corrected typo.

Round 2

Reviewer 2 Report

The manuscript became easier to follow and cohesive. The tables and figures are well integrated into the text. The conclusions are clear and deduced from the results.

Minor grammatical changes:

·         In line 186: between is repeated. Please, delete it.

·         In line 201: “CWP patients” at the end of the sentence feels redundant.

Author Response

The manuscript became easier to follow and cohesive. The tables and figures are well integrated into the text. The conclusions are clear and deduced from the results.

Minor grammatical changes:

  • In line 186: between is repeated. Please, delete it

-Thank you for the valuable comment, we delete it

  • In line 201: “CWP patients” at the end of the sentence feels redundant.

-Thank you for the valuable comment, we revised it to reduce redundancy.

Furthermore, we checked the grammar of our abstract and contents.